# Radiomics Score Combined with ACR TI-RADS in Discriminating Benign and Malignant Thyroid Nodules Based on Ultrasound Images: A Retrospective Study

**DOI:** 10.3390/diagnostics11061011

**Published:** 2021-06-01

**Authors:** Peng Luo, Zheng Fang, Ping Zhang, Yang Yang, Hua Zhang, Lei Su, Zhigang Wang, Jianli Ren

**Affiliations:** 1Department of Ultrasound, The Second Affiliated Hospital of Chongqing Medical University, Chongqing 400010, China; luopengjjyy@163.com (P.L.); rainingzp@163.com (P.Z.); yangyang1983818@163.com (Y.Y.); zhanghua052243@126.com (H.Z.); nicolsue@163.com (L.S.); wzg62942443@163.com (Z.W.); 2Department of Radiology, The Second Affiliated Hospital of Chongqing Medical University, Chongqing 400010, China; wwwfans2000@hospital.cqmu.edu.cn

**Keywords:** ACR TI-RADS, radiomics, Rad-score, thyroid nodules, ultrasound

## Abstract

This study aimed to explore the ability of combination model of ultrasound radiomics score (Rad-score) and the thyroid imaging, reporting and data system by the American College of Radiology (ACR TI-RADS) in predicting benign and malignant thyroid nodules (TNs). Up to 286 radiomics features were extracted from ultrasound images of TNs. By using the lowest probability of classification error and average correlation coefficients (POE + ACC) and the least absolute shrinkage and selection operator (LASSO), we finally selected four features to establish Rad-score (Vertl-RLNonUni, Vertl-GLevNonU, WavEnLH-s4 and WavEnHL-s5). DeLong’s test and decision curve analysis (DCA) showed that the method of combining Rad-score and ACR TI-RADS had the best performance (the area under the receiver operating characteristic curve (AUC = 0.913 (95% confidence interval (CI), 0.881–0.939) and 0.899 (95%CI, 0.840–0.942) in the training group and verification group, respectively), followed by ACR TI-RADS (AUC = 0.898 (95%CI, 0.863–0.926) and 0.870 (95%CI, 0.806–0.919) in the training group and verification group, respectively), and followed by Rad-score (AUC = 0.750 (95%CI, 0.704–0.792) and 0.750 (95%CI, 0.672–0.817) in the training group and verification group, respectively). We concluded that the ability of ultrasound Rad-score to distinguish benign and malignant TNs was not as good as that of ACR TI-RADS, and the ability of the combination model of Rad-score and ACR TI-RADS to discriminate benign and malignant TNs was better than ACR TI-RADS or Rad-score alone. Ultrasound Rad-score might play a potential role in improving the differentiation of malignant TNs from benign TNs.

## 1. Introduction

Thyroid nodules (TNs) are common diseases in the endocrine system [1]. With the strengthening of people’s health awareness and the improvement of examination techniques, the detection rate of TNs is increasing year by year, and the incidence of TNs by high-frequency ultrasound among adults is 68% [2]. Large-scale statistics showed that the total prevalence rate of TNs in Beijing was 49.0%, and the age-standardized prevalence rate was 40.1%, which increased significantly with aging [3]. In addition, the mortality rate associated with thyroid cancer did not change significantly, but the detection rate of thyroid cancer increased substantially [4,5]. The prognosis and clinical treatment of patients with TNs are mainly related to their pathological states. The pathological states of TNs are usually obtained by fine-needle aspiration (FNA). High-frequency ultrasound is the most commonly used method for thyroid imaging examination. The American College of Radiology thyroid imaging, reporting and data system (ACR TI-RADS) is currently the most commonly used tool for the noninvasive risk stratification of TNs in clinical practice and describes the ultrasound manifestations of TNs from five aspects: composition, echogenicity, shape, margins and echogenic foci [6,7]. However, some malignant TNs show benign features in ultrasound images, such as wider-than-tall shape, smooth margins and no calcification [8]. ACR TI-RADS misjudges the malignant risk of these nodules and tends to make a judgment of low-grade risk stratification. However, these nodules have a certain degree of invasiveness, and this misjudgment can cause patients to miss the best time for treatment and even cause serious consequences. Therefore, the evaluation value of ACR TI-RADS for these kinds of TNs is limited.

Radiomics is a new medical image analysis method developed in recent years [9,10,11,12]. The relationship between medical images and pathology can be established by extracting a large number of quantitative features from medical images for analysis, which can serve as the basis of clinical decision-making, thus assisting in the accurate diagnosis and treatment. Recently, an increasing number of studies have found that the features obtained from medical images might be related to the histopathological states of diseases [10,13,14,15]. Then, radiomics could be used to stratify the grading of cancer and the risk of cancer cell metastasis [16,17]. More importantly, radiomics could predict protein expression, assisting molecular diagnosis [18], which has great potential in the context of precision medicine. Therefore, radiomics could provide a personalized and accurate medical method for cancer treatment.

A previous study demonstrated that radiomics had the potential to predict benign and malignant TNs [19]. For the visualization of TNs, high-frequency ultrasound is superior to other imaging methods, and it is the main means of thyroid imaging examination at present in clinical practice. The aim of this study was to investigate the feasibility of ultrasound radiomics in evaluating benign and malignant TNs and explore whether the combination of ultrasound radiomics and ACR TI-RADS could improve the ability to discriminate benign and malignant TNs.

## 2. Methods

### 2.1. Study Population

This study was a retrospective study that was approved by the Ethics Committee of our hospital, which waived the informed consent form of the patients. All data were obtained from the patients’ hospitalization history. The inclusion criteria were as follows: (1) patients who had thyroid ultrasound examination and TNs detected from January 2018 to December 2019 in our hospital; (2) patients with TNs who underwent thyroid surgery and obtained histopathological results; and (3) patients with TNs who did not receive any tumor-related treatment, such as radiotherapy or chemotherapy, before thyroid ultrasound examination and thyroid surgery. A total of 495 TNs with pathological results were included. The exclusion criteria were as follows: (1) the presence of shadow of calcification that blurred the boundary of the TNs and even blurred the structure of the TNs; (2) TNs that had controversial pathological results; (3) the marks in the ultrasound images of TNs were located in the interior of the nodules or at the edge of the nodules so that feature extraction was affected or the outline of the TNs was impacted. Then, we excluded 101 cases from the total cases. As a result, the study included 394 nodules from 394 patients (87 men and 307 women; mean age, 46.2 years; range, 20–83 years), of which there were 198 malignant nodules and 196 benign nodules. All 394 nodules were classified into the training group to establish models. Then, we randomly selected 150 nodules to form a verification group, of which 72 nodules were malignant, and 78 nodules were benign, and the patients had an average age of 45.9 years (range, 20–80 years) (Table 1).

### 2.2. TN Ultrasound Images

We searched the above patients’ information in the image storage system and obtained thyroid ultrasound images. All thyroid glands of the patients were examined using a Philips iU 22 system (Philips Healthcare, Eindhoven, The Netherlands), a Siemens Acuson S2000 ultrasound system (Siemens Medical Systems, Erlangen, Germany) or an Esaote MyLab 70 ultrasound (Esaote, Genova, Italy) equipped with a 5–15 MHz linear-array transducer. The two-dimensional (B-mode) ultrasound features of TNs were extracted according to the ACR TI-RADS standard [20]. We described the B-mode ultrasound features of each nodule, including composition, echogenicity, shape, margins and echogenic foci.

### 2.3. TNs Segmentation

The B-mode ultrasound images of TNs with the largest diameter were selected from the image storage system and then put into MaZda software version 4.6 (The Technical University of Lodz, Institute of Electronics) in BMP format [21]. The image mode and grayscale palette were adjusted to make the boundary of the TNs as clear as possible, and the boundary was drawn according to the region of interest (ROI). The ROIs were determined as follows: (1) the maximum cross-sections were drawn; (2) the range included the area of thyroid echo changes around the nodules.

### 2.4. Radiomics Feature Extraction, Dimension Reduction and Calculation of the Radiomics Score

A maximum of 286 features was extracted from the training group, including the 4 categories of histogram features, morphological features, texture features and wavelet features, and the conditions of the run-length matrix (RLM) bits/pixel, co-occurrence matrix (COM) bits/pixel, gradient and wavelets were 6, 6, 4 and 8, respectively. To determine which parameters would be most useful for predicting benign TNs and malignant TNs, the process of selecting the optimal features was divided into two steps. First, we used the lowest probability of classification error and average correlation coefficients (POE + ACC) to initially select the desired features, and then least absolute shrinkage and selection operator (LASSO) logistic regression using 10-fold cross-validation was used to make the final selection to calculate the radiomics score (Rad-score) [22,23,24].

### 2.5. Models

Binary logistic regression with backward stepwise selection was used to establish models based on the B-mode ultrasound features of ACR TI-RADS (Method1), the Rad-score (Method2) and the combination of the B-mode ultrasound features of ACR TI-RADS and the Rad-score (Method3). The sensitivity, specificity, accuracy, negative predictive value (NPV), positive predictive value (PPV), F1-score and area under the receiver operating characteristic curve (AUC) were used to quantify the performance of the models [25,26,27]. The verification group data were used to verify the effectiveness of the models. Decision curve analysis (DCA) was used to select the model that maximized patient benefits.

### 2.6. Statistical Analysis

All data were analyzed by SPSS Statistics software version 25.0 (IBM, Armonk, NY, USA), MedCalc software version 18.2.1 (MedCalc Software bvba, Ostend, Belgium), Empower (R) (www.empowerstats.com (accessed on 10 September 2020), X&Y Solutions, Inc., Boston, MA, USA) and R software (http://www.R-project.org (accessed on 10 September 2020)). Shapiro–Wilk tests were used to determine whether the continuous variables followed a normal distribution. The continuous variables conforming to a normal distribution were expressed as the mean ± standard deviation (SD), while those with a non-normal distribution were expressed as the median (interquartile range [IQR]). The Mann–Whitney U test was used to test the differences between the continuous variables, and the chi-square test was used to compare the differences between the categorical variables. DeLong’s test was used to compare the AUC values of the different prediction models [22,25]. A *p* < 0.05 was considered to be a significant difference.

## 3. Results

### 3.1. Clinical Factors of the Patients and the Model Based on ACR TI-RADS

Our study included 394 TNs from 394 patients, which were all included in the training group to build the models, of which 150 TNs were randomly selected as the verification group. In the training group, there were 198 TNs that were pathologically diagnosed as malignant nodules (papillary carcinoma, *n* = 196; medullary carcinoma, *n* = 2). There were 72 cases of malignant nodules in the verification group, all of which were papillary thyroid carcinomas (*p* = 0.638). There were 87 men and 307 women in the training group and 29 men and 121 women in the verification group (*p* = 0.484). The chi-square test showed that the differences in composition, echogenicity, shape, margins and echogenic foci between benign and malignant TNs were all statistically significant (*p* < 0.05) (Table 2). Therefore, the five parameters in the ACR TI-RADS lexicon were applied to establish the B-mode ultrasound model and the joint model. In addition, there was a significant difference in age between patients with benign and malignant nodules (*p* < 0.05), so after selecting the best model, we included age into the model and then compared the AUCs of both models. By using logistic regression, the five parameters of ACR TI-RADS in predicting malignant TNs had an AUC of 0.898 (95% confidence interval [CI], 0.863–0.926) in the training group and 0.870 (95% CI, 0.806–0.919) in the verification group. The sensitivity, specificity, accuracy, PPV, NPV and F1-score were 80.30%, 83.16%, 81.73%, 82.81%, 80.69% and 0.82 in the training group and 75.00%, 87.18%, 81.33%, 84.38%, 79.07% and 0.79 in the verification group, respectively (Table 3).

### 3.2. Feature Selection and Rad-Score Calculation

To avoid overfitting, first, we obtained ten features by using POE + ACC, and then we ran the LASSO algorithm for final dimensionality reduction and feature selection, obtaining the four most capable features to distinguish between benign and malignant TNs. We calculated the Rad-score based on these four selected features as follows (Lambda·1se[λ] = 0.0288, ten-fold cross-validation) (Figure 1):Rad-score = −0.00425 × Vertl-RLNonUni + 0.0162 × Vertl-GLevNonU − 0.00172 × WavEnLH-s4 + 0.00074 × WavEnHL-s5

There was a significant difference in the Rad-score between benign TNs and malignant TNs (−1.595 (−2.315, −1.096); −0.944 (−1.425, −0.694)) in the training group (*p* < 0.001). Then, in the verification group, the Rad-score also showed a statistically significant difference between benign TNs and malignant TNs (−1.617 (−2.372, −1.156); −0.979 (−1.516, −0.677)); *p* < 0.001) (Figure 2 and Figure 3) [28].

The prediction performance for malignant TNs of the Rad-score (AUC, 0.750 (95% CI, 0.704–0.792) in the training group and 0.750 (95% CI, 0.672–0.817) in the verification group) was not as good as that of the five parameters of ACR TI-RADS in the training group (*p* < 0.001) and verification group (*p* = 0.0059). However, we combined the Rad-score and the five parameters of ACR TI-RADS to establish a logistic regression model, which performed better than the previous two models in predicting benign and malignant TNs, with an AUC of 0.913 (95% CI, 0.881–0.939), a sensitivity of 87.37%, a specificity of 84.18%, an accuracy of 85.79%, a PPV of 84.80%, an NPV of 86.84% and an F1-score of 0.86 in the training group (*p* = 0.0346), which was confirmed in the verification group (*p* = 0.0202). Finally, we brought the factor of age into the third modeling method, which had no remarkable difference (*p* = 0.0675 in the training group and 0.1761 in the verification group) (Table 3 and Table 4, Figure 4).

### 3.3. DCA and the Construction of a Nomogram Based on the Rad-Score and the Five Parameters of ACR TI-RADS

Since the model including age did not significantly improve the ability to distinguish between benign and malignant TNs, age was not included in the following analyses. We implemented DCA as shown in Figure 5, which showed that if the threshold probability was >approximately 25%, the method that combined the Rad-score and the five parameters of ACR TI-RADS to distinguish between benign and malignant TNs would be more beneficial for patients than the Rad-score or the five parameters of ACR TI-RADS alone.

The nomogram, based on the Rad-score and the five parameters of ACR TI-RADS, and the calibration plot are shown in Figure 6.

## 4. Discussion

In this study, we established three models based on ACR TI-RADS, ultrasound radiomics and combining ACR TI-RADS and ultrasound radiomics to distinguish between benign and malignant TNs. We included 394 TNs and used the postoperative histopathology of TNs as the gold standard to evaluate the ability of the three models to distinguish between benign and malignant TNs. The most important finding of this study was that ultrasound radiomics could improve the ability of ACR TI-RADS to distinguish between benign and malignant TNs.

In the process of establishing the ultrasound radiomics model, we selected the four most capable features to distinguish between benign and malignant TNs: Vertl-RLNonUni, Vertl-GLevNonU, WavEnLH-s4 and WavEnHL-s5. Based on these four features, we developed a Rad-score for classifying benign and malignant TNs. In the training group and verification group, the Rad-scores of malignant TNs were higher than those of benign TNs.

Our study showed that although ultrasound radiomics had the potential to distinguish between benign and malignant TNs (AUC = 0.750 (95% CI, 0.704–0.792) and 0.750 (95% CI, 0.672–0.817), sensitivity = 73.74% and 68.06%, specificity = 61.22% and 67.95%, accuracy = 67.51% and 68.00%, PPV = 65.77% and 66.22%, NPV = 69.77% and 69.74% and F1-score = 0.70 and 0.67 in the training group and validation group, respectively), it was not as good as that shown in previous studies [19,29]. A possible reason for this result was that by using the two-step dimensionality reduction method (POE + ACC, LASSO), we obtained only the top four features that were most capable of distinguishing between benign and malignant TNs, which might cause excessive dimensionality reduction and filter out some features that were still able to distinguish between benign and malignant TNs. Furthermore, a similar study showed that the Rad-score (AUC, 0.921 in the training group and 0.931 in the validation group) performed significantly better than the ACR TI-RADS scores [30], which was contrary to our findings. In our study, the ability of the Rad-score to distinguish between benign and malignant TNs was weaker than that of ACR TI-RADS, which had AUCs of 0.898 (95% CI, 0.863–0.926) and 0.870 (95% CI, 0.806–0.919) in the training group and verification group, respectively. This may be because we did not calculate the total ACR TI-RADS score but brought the score of each parameter into the model separately, which might improve the ability to identify malignant TNs.

Then, we combined the Rad-score and ACR TI-RADS to establish a logistic regression model, which had the best ability to distinguish between benign and malignant TNs, with an AUC of 0.913 (95% CI, 0.881–0.939) in the training group and 0.899 (95% CI, 0.840–0.942) in the verification group. Furthermore, DCA showed that the combined model based on the Rad-score and ACR TI-RADS to predict malignant TNs was more beneficial to patients than the Rad-score model or the ACR TI-RADS model. The calibration curve of the combination model of the Rad-score and ACR TI-RADS showed that the mean absolute error was only 0.024. Therefore, the joint model of the Rad-score and ACR TI-RADS was superior to the other two models.

Why was it that the ability of the Rad-score alone to identify malignant TNs was not as good as that of ACR TI-RADS, but when combined with ACR TI-RADS, it could improve the ability to distinguish between benign and malignant TNs? We reviewed previous studies and found that some studies had suggested that ACR TI-RADS performed well in the risk stratification of TNs, and its AUC could reach up to 0.91 [19,30]. In addition, artificial intelligence-assisted TI-RADS performed better and could improve the diagnostic performance of junior radiologists [31,32]. Hypoechogenicity and microcalcification in ACR TI-RADS, especially microcalcification, were very useful criteria for predicting malignancy of TNs. If vascularity was added, excellent specificity could be achieved (99.6%, 95% CI (97.6%–99.9%)) [8]. However, the ultrasonographic features of some malignant TNs were similar to those of benign TNs, such as wider-than-tall shape, regular and smooth borders or margins and no microcalcification [8]. We determined that it was possible that the Rad-score had merely identified this kind of malignant TN with benign ultrasound features, which could improve the ability of ACR TI-RADS to recognize malignant thyroid lesions in the combination model, including the Rad-score and ACR TI-RADS. In addition, radiomics extracted the internal information of TNs ultrasound images at a microscopic level invisible to the naked eye to distinguish between benign and malignant TNs, while ACR TI-RADS evaluated TNs under macroscopic conditions visible to the naked eye. Because the two methods predicted benign and malignant TNs from two different aspects, when we combined the two methods, the complementary effect of radiomics and ACR TI-RADS improved the performance.

In addition, we combined two different dimensionality reduction methods to select the optimal features, which was different from that in previous studies, so whether this method could improve the efficiency of screening features needs to be confirmed by subsequent studies.

This study had several limitations. Firstly, the method of radiomics in this study was based on ultrasound images, but in the process of ultrasound image generation, operators needed to constantly adjust parameters to achieve the best image quality, and radiomics not only required good image quality but also needed to maintain the consistency of imaging parameters. Therefore, future studies need to study the corresponding imaging rules to achieve homogeneity in ultrasound images. Secondly, of the cases in this study, the vast majority of cases were papillary thyroid carcinoma, a few were medullary thyroid carcinoma, and there was no follicular thyroid carcinoma (FTC). In addition, ACR TI-RADS has limited usefulness in identifying FTC because FTC and follicular adenoma have similar ultrasound findings [33]. Therefore, whether ultrasound Rad-score combined with ACR TI-RADS can improve the ability to identify such nodules needs further research. Lastly, since this study did not consider the serum-thyroid stimulating hormone level or scintigraphy images of the contained nodules, and ACR TI-RADS may prompt a large number of patients with hot nodules to undergo inappropriate FNA [34], whether the combined effect of ACR TI-RADS and ultrasound Rad-score can reduce the unnecessary FNA ratio of hot nodules needs more research to explore.

## 5. Conclusions

Using the Rad-score alone to distinguish between benign and malignant TNs did not perform as well as ACR TI-RADS, but the ability of the combined model was better than that of the individual models. Therefore, the Rad-score may play a potential role in improving the ability of ACR TI-RADS to distinguish malignant TNs from benign TNs in clinical practice.

## Figures and Tables

**Figure 1 diagnostics-11-01011-f001:**
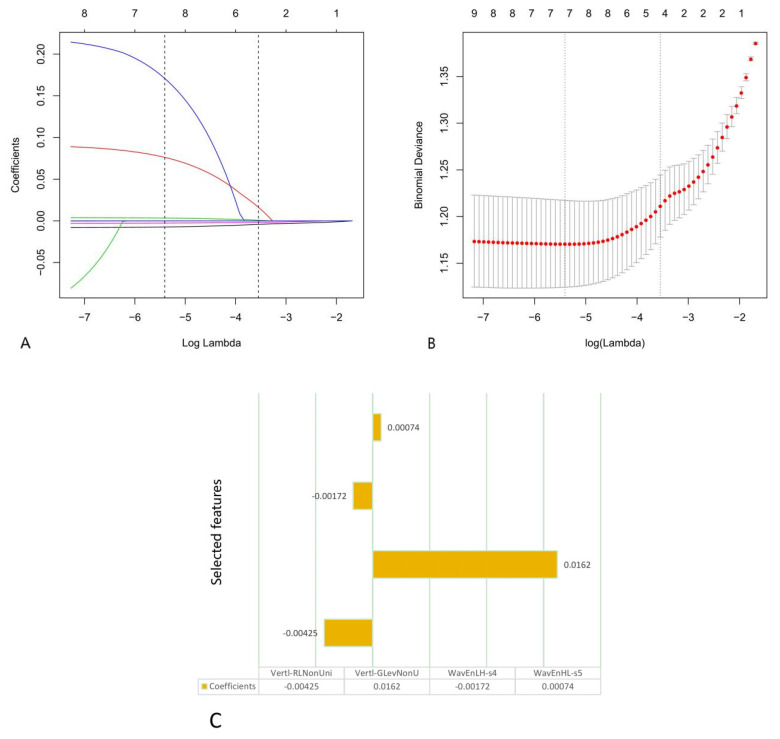
The second step of features selection using the least absolute shrinkage and selection operator (LASSO) logistic regression from ten features selected by the lowest probability of classification error and average correlation coefficients (POE + ACC). (**A**,**B**) Ten-fold cross-validation based on 1se was used to select the tuning parameter (λ) in the LASSO regression model. An optimal λ value of 0.0288 was selected. (**C**) The four selected features and coefficients of the Rad-score were indicated by the *y*-axis and *x*-axis, respectively.

**Figure 2 diagnostics-11-01011-f002:**
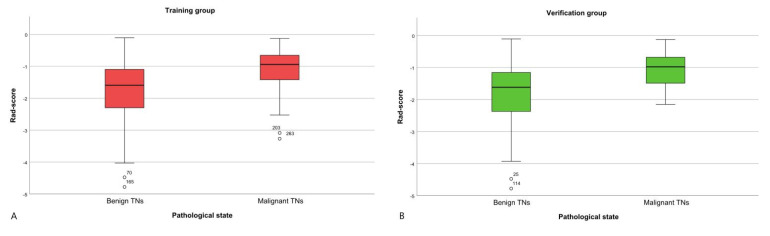
The box-and-whisker plot of the Rad-scores in the training and verification groups. (**A**) Training group. (**B**) Verification group.

**Figure 3 diagnostics-11-01011-f003:**
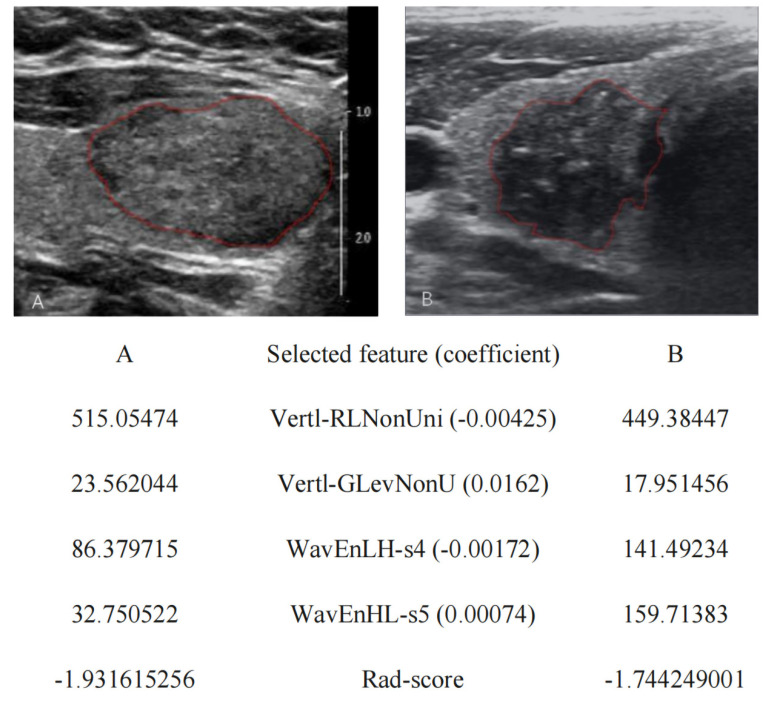
Selected ultrasound radiomics features and radiomics score (Rad-score) of two representative patients. (**A**) A benign thyroid nodule in a 47-year-old woman confirmed by postoperative histopathology was described according to the thyroid imaging, reporting and data system by the American College of Radiology (ACR TI-RADS) as solid (2 points), isoechoic (1 point), wider-than-tall (0 point), blurry margins (0 point) and no calcification (0 point). The Rad-score was −1.931615256; (**B**) A malignant thyroid nodule in a 47-year-old man confirmed by postoperative histopathology was described according to the ACR TI-RADS as solid (2 points), hypoechoic (2 points), taller-than-wide (3 points), irregular margins (2 points) and microcalcification (3 points). The Rad-score was −1.744249001.

**Figure 4 diagnostics-11-01011-f004:**
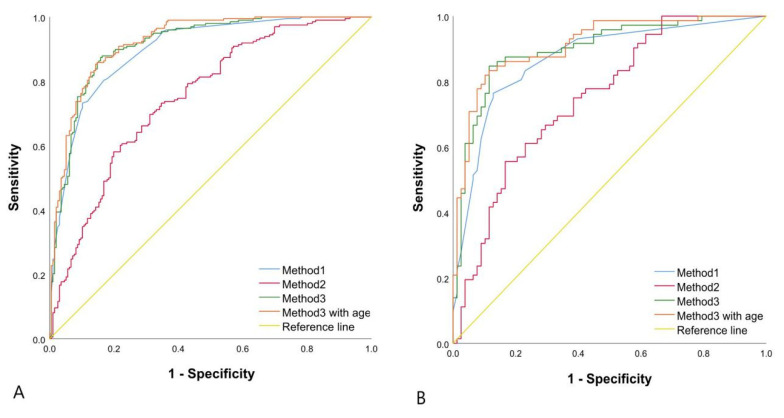
Binary logistic regression ROC curve for discriminating benign and malignant TNs. (**A**) The training group: Method1 AUC of 0.898 (95%CI, 0.863–0.926); Method2 AUC of 0.750 (95%CI, 0.704–0.792); Method3 AUC of 0.913 (95%CI, 0.881–0.939). (**B**) The verification group: Method1 AUC of 0.870 (95%CI, 0.806–0.919); Method2 AUC of 0.750 (95%CI, 0.672–0.817); Method3 AUC of 0.899 (95%CI, 0.840–0.942). Method3 with age in A and B represented the training group and verification group of Method3 of joining age, respectively. The AUC of Method3 with age in A and B was 0.923 (95%CI, 0.892–0.947) and 0.912 (95%CI, 0.855–0.952), respectively.

**Figure 5 diagnostics-11-01011-f005:**
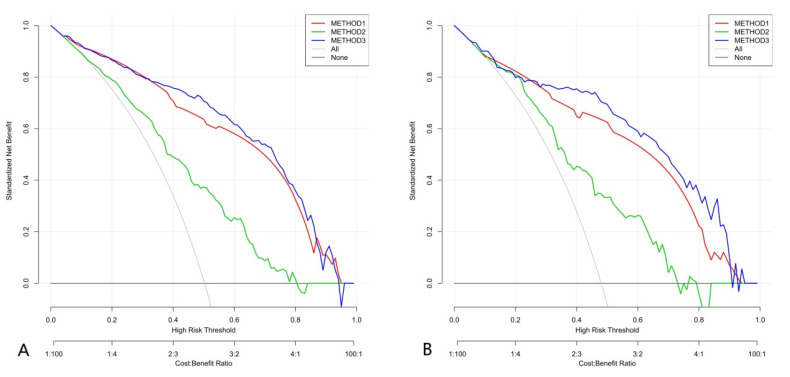
Decision curve analysis (DCA) of three methods. Net benefit (*y*-axis) = true positive rate—(false positive rate × weighting factor), weighting factor = threshold probability/(1-threshold probability). The DCA showed that if the threshold probability was >about 25%, the method that combined the Rad-score and the five parameters of ACR TI-RADS to distinguish between benign and malignant TNs would be more beneficial for patients than the Rad-score or the five parameters of ACR TI-RADS alone. (**A**) Training group. (**B**) Verification group.

**Figure 6 diagnostics-11-01011-f006:**
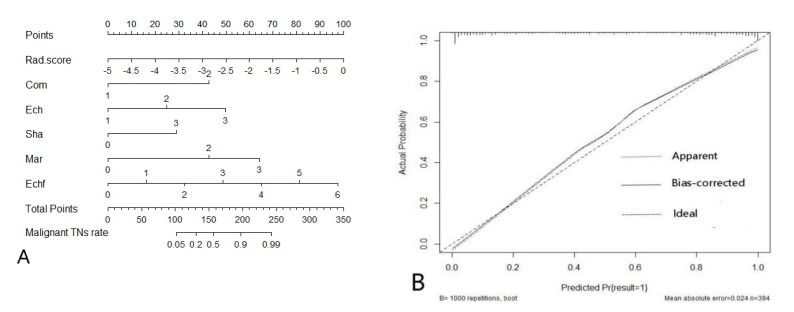
(**A**) The nomogram combining ultrasound radiomics score (Rad-score) and the five parameters of ACR TI-RADS developed in the training group. (**B**) The calibration curve showed a mean absolute error of 0.024. result = 1: The pathological state of a nodule was malignant.

**Table 1 diagnostics-11-01011-t001:** The characteristics of the 394 patients included in the study.

Training Group		Verification Group	
Benign (196)	Malignant (198)	Benign (78)	Malignant (72)
44	43	14	15
152	155	64	57
50.1 (20–80)	42.2 (20–83)	49.1 (20–80)	42.4 (20–79)

**Table 2 diagnostics-11-01011-t002:** Parameters of patients in benign and malignant TNs included in the training group.

Parameters	Training Group (394)		*p* Value
	Benign (196)	Malignant (198)	
Age (yr)	49.0 (43.3, 56.0)	42.0 (31.8, 51.0)	<0.001 *
Gender			
Male	44	43	0.861
Female	152	155	
Composition			
Cystic, dominantly cystic or spongiform	0	0	-
Cystic-solid mixture	53	1	<0.001 *
Solid, dominantly solid	143	197	
Echogenicity			<0.001 *
Anechoic			
Hyperechoic or isoechoic	148	45	
Hypoechoic	45	138	
Very hypoechoic	3	15	
Shape			
Wider-than-tall	176	97	<0.001 *
Taller-than-wide	20	101	
Margins			<0.001 *
Smooth or blurry	179	106	
Lobed or irregular	17	69	
Extrathyroid extension	0	23	
Echogenic foci †			<0.001 *
0	173	124	
1	4	16	
2	2	4	
3	17	48	
≥4	0	6	

Not all age groups were normally distributed, so ages were shown as median (interquartile range [IQR]). * *p*-value < 0.05. † 0, 1, 2, 3, ≥4 represented no calcification or with comet tail sign, coarse calcification, perinodular annular calcification, microcalcification or more than two types of calcification except comet tail sign whose combined score was greater than or equal to 4, respectively.

**Table 3 diagnostics-11-01011-t003:** Diagnostic performances of models.

	Training Group							Verification Group						
	AUC	SEN (%)	SPE (%)	ACC (%)	PPV (%)	NPV (%)	F1-Score	AUC	SEN (%)	SPE (%)	ACC (%)	PPV (%)	NPV (%)	F1-Score
Method1	0.898	80.30	83.16	81.73	82.81	80.69	0.82	0.870	75.00	87.18	81.33	84.38	79.07	0.79
Method2	0.750	73.74	61.22	67.51	65.77	69.77	0.70	0.750	68.06	67.95	68.00	66.22	69.74	0.67
Method3	0.913	87.37	84.18	85.79	84.80	86.84	0.86	0.899	80.56	88.46	84.67	86.57	83.13	0.83
Method3 with ages	0.923	85.86	84.69	85.28	85.00	85.57	0.85	0.912	83.33	87.18	85.33	85.71	85.00	0.85

SEN, sensitivity; SPE, specificity; ACC, accuracy; PPV, positive predictive value; and NPV, negative predictive value. Method1, Method2, Method3 represented the modeling methods of the five parameters of ACR TI-RADS, Rad-score and the combination of Rad-score and the five parameters of ACR TI-RADS, respectively.

**Table 4 diagnostics-11-01011-t004:** The differences of AUC between different modeling methods analyzed by DeLong’s test.

	Training Group		Verification Group	
	Difference between Areas	*p* Value	Difference between Areas	*p* Value
Method1 vs. Method2	0.148	<0.001	0.120	0.0059
Method1 vs. Method3	0.015	0.0346	0.029	0.0202
Method2 vs. Method3	0.163	<0.001	0.149	<0.001
Method3 vs. Method3 with age	0.010	0.0675	0.013	0.1761

## Data Availability

Raw data can be shared from the first author if there is a reasonable request.

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
