# Peer review of "Radiomics Score Combined with ACR TI-RADS in Discriminating Benign and Malignant Thyroid Nodules Based on Ultrasound Images: A Retrospective Study"

_diagnostics, 2021, doi:10.3390/diagnostics11061011_

Round 1
Reviewer 1 Report
This study will be helpful for clinical practice, when physicians could dispose of a radiomics software. This study is well-written and conclusions are in agreement with the results. Also, tables and figures are representative.
I only have some minor remarks:
1. Please shorten the introduction (for example you can delete the explanations of how ACR works and its details).
2. Since all the included malignant nodules are papillary thyroid cancer, the results only apply to this hystotype. Therefore, you can add this issue as a limitation of your work and in this regard i recommend to cite the following article: Castellana M. et al. "Can ultrasound systems for risk stratification of thyroid nodules identify follicular carcinoma? Cancer Cytopathol 2020"
3. Since you do not consider serum TSH levels or scintigraphy images of the included nodules, we can not known how your results apply to hot nodules. Therefore, you can add this issue as a limitation and in this regard i recommend to cite the following article: Castellana M. et al. "Ultrasound systems for risk stratification of thyroid nodules prompt inappropriate biopsy in autonomously functioning thyroid nodules" Clinical Endocrinol (Oxf) 2020.
Reviewer 2 Report
The paragraph 2.4 must be redrafted/supplemented. In the current form, the description of obtaining data for radionomics score is unintelligible. Moreover, the authors do not explain how the Rad-score is calculated. There is a noticeable disproportion between the description of ACR TI-RADS vs Rad-score. Before publishing, it is necessary to supplement the manuscript with examples of ACR TI-RADS and Rad-score calculation, of both benign and malignant TNs. Figure 1 is unnecessary.
Author Response
Response to Reviewer 2 Comments
Thanks to reviewer 2 for the suggestions for the manuscript, we revised the manuscript as follows based on the comments of reviewer 2.
Point 1: The paragraph 2.4 must be redrafted/supplemented. In the current form, the description of obtaining data for radionomics score is unintelligible. Moreover, the authors do not explain how the Rad-score is calculated.
Response 1: Regarding the calculation of Rad-score, we have the following explanation, in this study, we extracted 286 radiomics features of each thyroid nodule. In order to avoid overfitting, it was necessary to select the most capable of distinguishing benign and malignant thyroid nodules from these 286 features. When the optimal features were selected by LASSO, the coefficient of each optimal feature would be generated. Based on these optimal features, Rad-score can be calculated (Rad-score = coefficient 1 × feature 1 + coefficient 2 × feature 2 + coefficient 3 × feature 3 +...), for example, we selected 4 features that were most capable of distinguishing benign and malignant thyroid nodules, their coefficients were -0.00425, 0.0162, -0.00172, 0.00074, respectively. Therefore, Rad-score =-0.00425 × feature 1 + 0.0162 × feature 2 - 0.00172 × feature 3 + 0.00074 × feature 4. The details are described in paragraph 3.2.
Point 2:There is a noticeable disproportion between the description of ACR TI-RADS vs Rad-score. Before publishing, it is necessary to supplement the manuscript with examples of ACR TI-RADS and Rad-score calculation, of both benign and malignant TNs.
Response 2: We have supplemented the manuscript with examples of ACR TI-RADS and Rad-score calculation, of both benign and malignant TNs in Figure 3 (A: a benign thyroid nodule. B: a malignant thyroid nodule).
Point 3: Figure 1 is unnecessary.
Response 3: We have deleted the Figure 1.

Round 2
Reviewer 2 Report
The authors have addressed most of my earlier comments. The authors have added new material to the manuscript which has generated additional comments, however
Line 363 „Lastly, since this study did not consider the serum thyroid stimulating hormone level or scintigraphy images of the contained nodules [34], further researches are needed to determine whether the conclusions of this study are applicable to hot nodules.”
The inclusion criteria in this research were among others: patients with TNs who underwent thyroid surgery and obtained histopathological results. This condition is not satisfied in patient's group with hot nodules. What the authors have in mind?
Author Response
Response to Reviewer 2 Comments (Round 2)
Point: Line 363 “Lastly, since this study did not consider the serum thyroid stimulating hormone level or scintigraphy images of the contained nodules [34], further researches are needed to determine whether the conclusions of this study are applicable to hot nodules.”
The inclusion criteria in this research were among others: patients with TNs who underwent thyroid surgery and obtained histopathological results. This condition is not satisfied in patient's group with hot nodules. What the authors have in mind?
Response: Thank you for your letter and for the reviewer’s comment concerning our manuscript. Regarding the content of line 363, we have the following explanation: what we originally wanted to express is that ACR TI-RADS may prompt a large number of patients with hot nodules to undergo inappropriate FNA, and for the assessment of benign and malignant thyroid nodules, since the combined effect of ACR TI-RADS and ultrasound Rad-score has a good performance. So, whether the combined effect of ACR TI-RADS and ultrasound Rad-score can reduce the unnecessary FNA ratio of hot nodules needs more research and exploration. We have modified this sentence to “Lastly, since this study did not consider the serum thyroid stimulating hormone level or scintigraphy images of the contained nodules, and ACR TI-RADS may prompt a large number of patients with hot nodules to undergo inappropriate FNA, whether the combined effect of ACR TI-RADS and ultrasound Rad-score can reduce the unnecessary FNA ratio of hot nodules needs more researches to explore”.
Special thanks to you for your good comments.
Round 3
Reviewer 2 Report
Thank you